# A Case of William’s Syndrome in a Ugandan Child: A Feasible Diagnosis Even in a Low-Resource Setting

**DOI:** 10.3390/children8121192

**Published:** 2021-12-16

**Authors:** Massimo Mapelli, Paola Zagni, Valeria Calbi, Aliku Twalib, Roberto Ferrara, Piergiuseppe Agostoni

**Affiliations:** 1Centro Cardiologico Monzino, Scientific Institute for Research, Hospitalization and Healthcare (IRCCS), 20138 Milan, Italy; piergiuseppe.agostoni@ccfm.it; 2Department of Clinical Sciences and Community Health, Cardiovascular Section, University of Milan, 20122 Milan, Italy; 3Terapia Intensiva Neonatale, Ospedale Fatebenefratelli P.O. Macedonio Melloni, Via Macedonio Melloni 52, 20129 Milan, Italy; paola.zagni@gmail.com; 4San Raffaele Telethon Institute for Gene Therapy (SR-TIGET), IRCCS San Raffaele Scientific Institute, Via Olgettina, 60, 20132 Milan, Italy; calbi1.valeria@hsr.it; 5Pediatric Immunohematology Unit and BMT Program, IRCCS San Raffaele Scientific Institute, Via Olgettina, 60, 20132 Milan, Italy; 6Division of Paediatric Cardiology Uganda Heart Institute, Mulago Hospital and Complex, Kampala P.O. Box 37392, Uganda; aliku90@yahoo.com; 7Medical Oncology Department, Fondazione IRCCS Istituto Nazionale dei Tumori, 20133 Milan, Italy; roberto.ferrara@istitutotumori.mi.it; 8Department of Research, Molecular Immunology Unit, Fondazione IRCCS Istituto Nazionale dei Tumori, 20133 Milan, Italy

**Keywords:** Williams’ syndrome, supravalvular aortic stenosis, congenital heart diseases, Williams’ syndrome in sub-Saharan Africa, echocardiography

## Abstract

Background: Williams–Beuren syndrome (WS) is a rare, complex, congenital developmental disorder including cardiovascular manifestations, intellectual disability and a peculiar cognitive and behavior profile. Supravalvular aortic stenosis (SVAS) is the most frequent cardiovascular abnormality in WS children. Data on WS patients in sub-Saharan Africa are scarce. A genetic study is usually required for a definite diagnosis, but genetic testing is often unavailable in developing countries and the combination of a typical clinical phenotype and echocardiographic profile helps to confirm the diagnosis. Case Report: We report the case of a 5-year-old Ugandan child admitted to a large no profit hospital after he was initially managed as a case of infective endocarditis. A physical examination revealed the typical features of WS. A cardiac echo showed severe SVAS (peak gradient 80 mmHg) with a normal anatomy and function of the aortic valve and mild valvular pulmonary stenosis. The child also had a moderate intellectual disability and a characteristic facies consistent with WS. Conclusion: We present the first reported case of WS in Uganda. Cardiac echo and a characteristic clinical picture could be enough to exclude more common causes of heart failure (i.e., rheumatic heart disease) and to make the diagnosis even when specific genetic tests are not available.

## 1. Case Report

We report the case of a 5-year-old Ugandan child (O.B.) who was referred to a large nonprofit Ugandan hospital (Lacor Hospital, Gulu, North Uganda) for dyspnea and fever for two weeks with a referral diagnosis of “Rheumatic heart disease with superimposed endocarditis resulting in severe aortic stenosis”. On admission, the patient was febrile (37.5 °C) and sick looking with a dry cough and mild respiratory distress. On examination, the chest was clear with no signs of pulmonary congestion. Remarkably, the child had a characteristic facies with small chin, eyes puffiness, epicanthal folds and a long philtrum (Figure 1). The weight was 13.5 Kg and height was 102 cm. No peripheral edema was noted. The patient was tachycardic with a regular heart rate of 115 beats per minute. The child’s blood pressure was 120/70 mmHg, which indicates hypertension for their age. He had a grade 4/6 ejection systolic murmur aortic valve area, radiating to the neck. The abdomen was soft without palpable masses.

We also noted features of moderate intellectual disability with poor short-term memory and a very limited vocabulary. He was also irritable and appeared restless at times. In addition, the patient showed signs of behavior uncommon among his peers. In particular, even though he was admitted to a hospital, he was remarkably sociable and prone to laughter and joking. Conversely, at times he would become frightened and burst into tears in response to modest stimuli (e.g., if a child near him raised his voice). In addition, the father reported that his son was used to react abnormally to intense noisy stimuli, such as a thunder during a thunderstorm, although this was a very frequent occurrence given the equatorial climate of the region (hypersensitivity to sounds). The rest of his neurologic physical exam was unremarkable. Specific neuropsychological tests were not conducted because these were not available in this setting.

A blood culture was negative and other laboratory findings, including full blood count, renal and liver function and electrolytes, were unremarkable. The ECG showed a sinus rhythm with LVH (Figure 2). The cardiac echo showed a mild thickening of the aortic valve cusps without images suspected for vegetations and with a normal systolic motion. There was a discrete narrowing of ascending aorta (hourglass deformity) just above the sino-tubular junction. The ascending aorta diameter was 11.8 cm (Z-SCORE = 3.2 SD) (Figure 3). The color-Doppler mapping showed a mosaic color pattern along the aortic root and proximal ascending aorta, while the continuous wave Doppler mode demonstrated a significant gradient (80 mmHg with a peak velocity of 4.5 m/s) consistent with a severe aortic stenosis (Figure 4a,b). Left ventricle concentric hypertrophy was also noted: LVPWDd = 0.93 cm (Zscore = 3.98 SD). There was also turbulent flow across the pulmonary valve with mild valvular pulmonary stenosis (pressure gradient of 36 mmHg; peak velocity of 3 m/s). There was a post-stenotic dilatation of the main pulmonary artery. No other cardiac abnormalities were not noted. The aortic arch was normal (Figure 5)

According to the echo findings and the clinical picture, a diagnosis of supravalvular aortic stenosis (SVAS) in Williams’ syndrome was made. The child was treated with HF medications, antibiotics for pneumonia and referred to a tertiary center with cardiac surgery at the capital for further management.

## 2. Discussion

In 1961, Williams described a group of four children with supravalvular aortic stenosis (SVAS) and a peculiar intellectual disability [1]. Williams–Beuren syndrome (WS) is a rare (incidence is estimated at 1:25,000 live births) and complex developmental disorder including cardiovascular manifestations. Moreover, intellectual disability with a peculiar cognitive and/or behavior profile, a characteristic facies and occasional hypercalcemia are also described [2].

From a genetic point of view, it is associated with a microdeletion in the 7q11.23 chromosomal region, which encompass the elastin gene. However, the proper pathogenic mechanism leading to the extensive vasculopathy is not entirely clear. In clinical studies, it has been shown that after 1 year, pulmonary arterial stenosis (PAS) tends to decline and SVAS worsens [3,4,5]. The difficulty in the growth of the sino-tubular junction has been presented as a possible pathogenetic element to explain the progression of aortic disease [5,6].

The definite diagnosis should be made by the clinical picture assessed by a medical geneticist together with the demonstration of the typical elastin gene hemizygosity (assessed by FISH); however, genetic testing is often unavailable in developing countries. However, the combination of a typical clinical phenotype and echocardiographic profile could help to confirm the diagnosis.

In addition to the ascending aorta, stenosis may also occasionally occur in the aortic arc, the carotid and innominate arteries. Cardiovascular symptoms were previously reported in 47% of patients, while in a large series 77% were found to have a structural heart defect [7]. SVAS was described as the most common diagnosis among children (79% of the cases). PAS is also common (41% of the subjects with cardiovascular manifestations). As a matter of fact, in our patient we observed a severe SVAS (peak gradient 80 mmHg) with normal anatomy and function of the aortic valve and mild valvular pulmonary stenosis, which are both fairly rare cardiovascular manifestations and compatible with the WS diagnosis.

In WS subjects, a characteristic so-called “elfin face” is also typical, consisting of a prominent metopic suture, small chin, sunken nasal bridge, eye puffiness, wide mouth, and prominent lower lip. This was also noted in our patient (Figure 1).

During the neonatal period, cardiac manifestations are common in WS children and might help in making an early diagnosis when other features can remain unrecognized. On the other hand, a considerable percentage of the WS-associated cardiovascular problems may not manifest until adult age due to the fact that symptoms might be missing or not specific, delaying or preventing dedicated diagnostic procedures and proper treatments. A detailed cardiac evaluation must be performed in all WS patients due to the high prevalence of cardiovascular abnormalities [8]. In addition to a diagnostic point of view, this is even more relevant to prognosis since when cardiac interventions are possible, the prognosis was relatively good and operative mortality low [7].

## 3. Conclusions

We present the first reported case of Williams’ syndrome in Uganda. Cardiac echo and a characteristic clinical picture could be enough to exclude more common causes of heart failure (i.e., rheumatic heart disease) and to make a diagnosis even when specific genetic tests are not available.

## Figures and Tables

**Figure 1 children-08-01192-f001:**
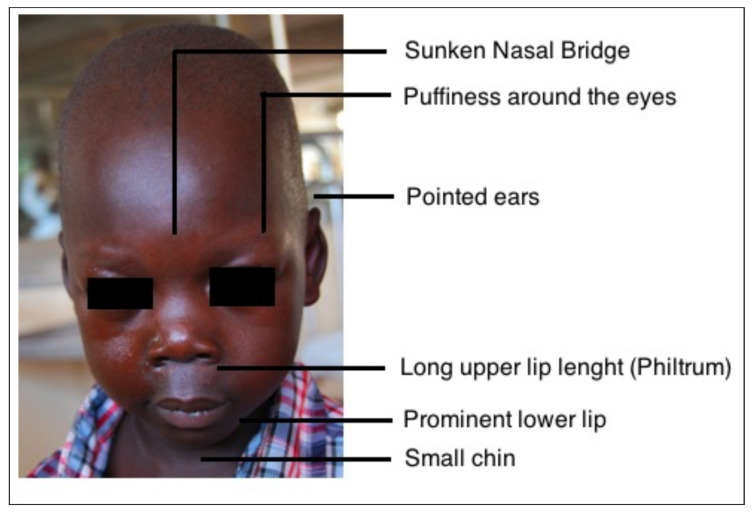
Typical “elfin face”. Williams’ syndrome children show few peculiar facial features who can suggest the diagnosis.

**Figure 2 children-08-01192-f002:**
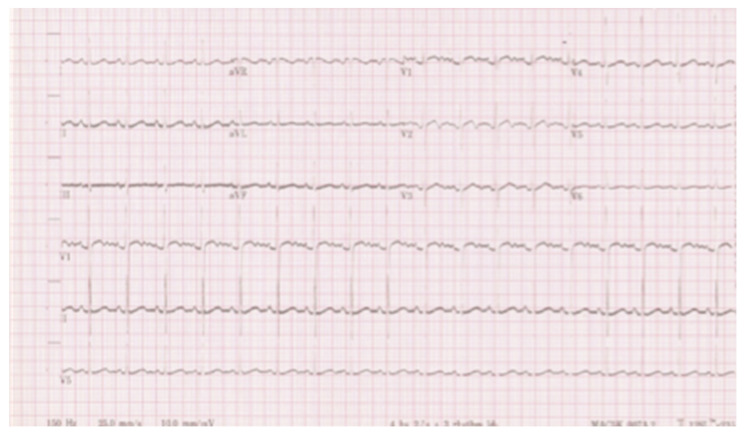
ECG. ECG showed sinus tachycardia with signs of left ventricle hypertrophy.

**Figure 3 children-08-01192-f003:**
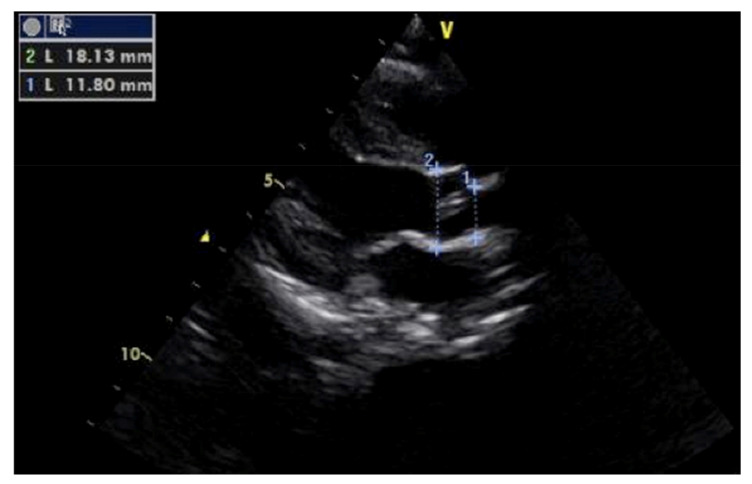
Proximal ascending aorta narrowing. Cardiac echo (long axis view) showed a discrete narrowing of ascending aorta (hourglass deformity) just above the sino-tubular junction. Ascending aorta diameter was 11.8 cm (Z-SCORE = 3.2 SD).

**Figure 4 children-08-01192-f004:**
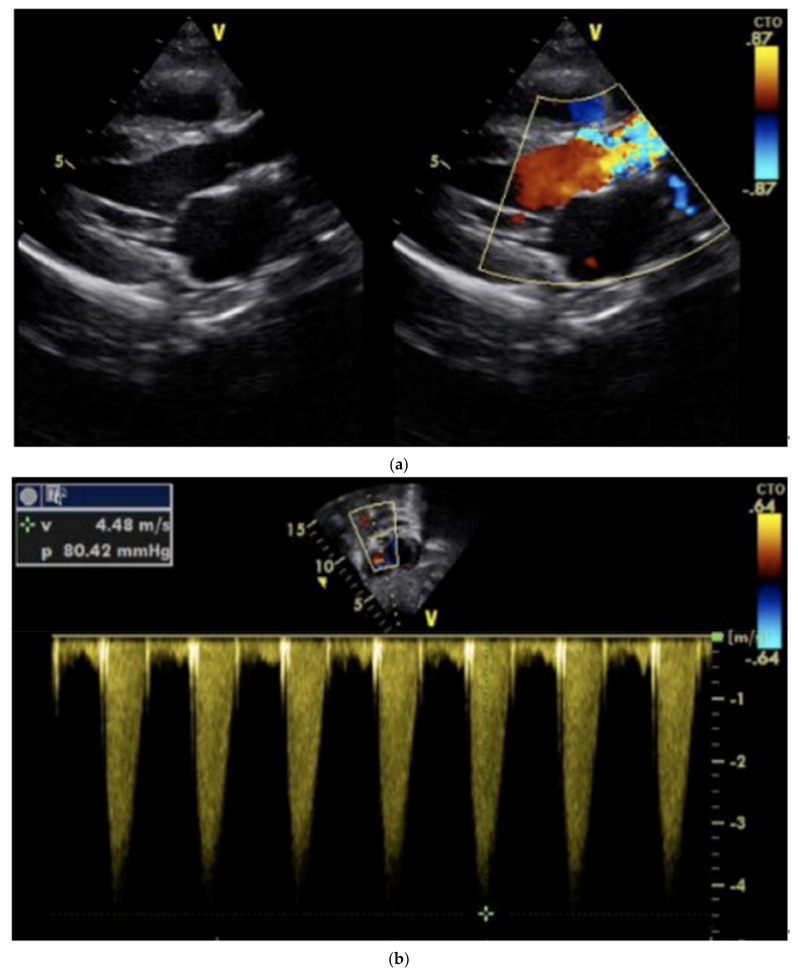
(**a**) Severe supravalvular aortic stenosis. Color-Doppler mapping showed a mosaic color pattern along the aortic root and proximal ascending aorta. (**b**) The continuous-wave Doppler mode demonstrated a significant gradient (80 mmHg with a peak velocity of 4.5 m/s) consistent with a severe aortic stenosis.

**Figure 5 children-08-01192-f005:**
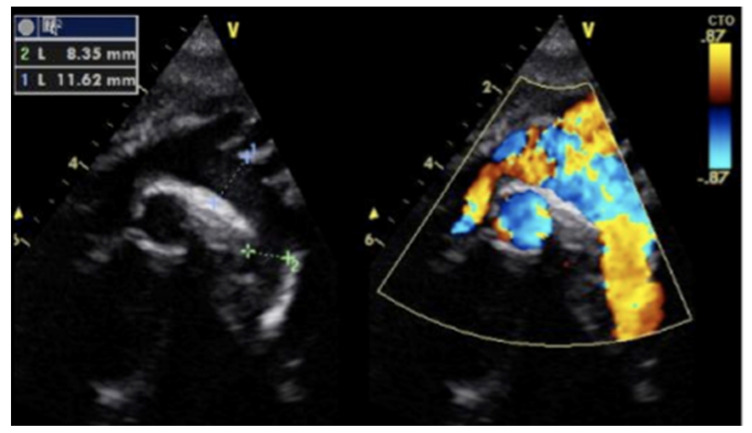
Normal aortic arch, proximal narrowing. The aortic arch was normal. A proximal narrowing of ascending aorta was noted also from the suprasternal projection.

## Data Availability

No database is available due to the nature of the study (single patient observational case report).

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
