# Peer review of "A Case of William’s Syndrome in a Ugandan Child: A Feasible Diagnosis Even in a Low-Resource Setting"

_children, 2021, doi:10.3390/children8121192_

Round 1

Reviewer 1 Report

Thank You for the opportunity to review an excellent manuscript. The manuscript describes a case of a very rare disease, for the first time diagnosed in Uganda. The case is described in very good detail with excellent documentation of the findings. This is the first case description in an Ugandan child in a very rare disease. The disease has been described in children of other ethnicities. To document the findings in this case is new and important. The paper is excellently written. The text is clear, interesting and easy to read.  The consclusions are consistent with the evidence presented. As this is a case description, this point is irrelevant

Reviewer 2 Report

This manuscript describing the first reported case of WS in Uganda provides a detailed medical description of the cardiac abnormalities characteristic of the disorder that can be used in lieu of genetic diagnosis. This is significant, as a standardized medical description for the diagnosis of WS is essential in order to provide proper medical care for WS individuals in regions without access to costly genetic testing. Overall, this case report achieves this aim, with a few areas of improvement:

  • in addition to the cardiac abnormalities and facial features, the unique and well-defined behavioral profile of WS (hypersociability, anxiety, etc) is an important part of diagnosis. However, no description of the behavioral profile is mentioned in this case report. While the report mentions that cognitive testing was not accessible, at the very least a more detailed background about typical WS behavioral profile, as well as qualitative observations about the patient's behavior, is needed.
  • Please note that "mental retardation" is no longer considered appropriate terminology, in the medical literature. This must be changed to "intellectual disability"
